

# Development of genomic simple sequence repeat markers for *Glycyrrhiza lepidota* and cross-amplification of other *Glycyrrhiza* species

Jun Hyoung Bang[1], Chi Eun Hong[2], Sebastin Raveendar[3], Kyong Hwan Bang[2], Kyung Ho Ma[2], Soon Wook Kwon[4], Hojin Ryu[5], Ick Hyun Jo[2] and Jong-Wook Chung[1]

[1] Department of Industrial Plant Science and Technology, Chungbuk National University, Cheongju, South Korea
[2] Department of Herbal Crop Research, National Institute of Horticultural and Herbal Science, Eumseong, South Korea
[3] National Agrobiodiversity Center, National Institute of Agricultural Science, Jeonju, South Korea
[4] Department of Plant Bioscience, Pusan National University, Miryang, South Korea
[5] Department of Biology, Chungbuk National University, Cheongju, South Korea

Corresponding authors
Ick Hyun Jo, intron@korea.kr
Jong-Wook Chung,
jakdukong@gmail.com,
jwchung73@chungbuk.ac.kr

## ABSTRACT

**Background**. Licorice (*Glycyrrhiza* spp. L.) is used as a natural sweetener and medicinal herb in European and Asian countries. Molecular studies have been conducted to find differences between wild and cultivated species because most wild species are highly resistant to abiotic and biotic stresses compared with their cultivated species. However, few molecular markers have been developed for studying the genetic diversity and population structure of licorice species and to identify differences between cultivars. Thus, the present study aimed to develop a set of genomic simple sequence repeat (SSR) markers for molecular studies of these species.

**Methods**. In the present study, we developed polymorphic SSR markers based on whole-genomesequence data of *Glycyrrhiza lepidota*. Then, based on the sequence information, the polymorphic SSR markers were developed. The SSR markers were applied to 23 *Glycyrrhiza* individual plants. We also evaluated the phylogenetic relationships and interspecies transferability among samples.

**Results**. The genetic diversity analysis using these markers identified 2–23 alleles, and the major allele frequency, observed heterozygosity, genetic diversity, and polymorphism information content were 0.11–0.91, 0–0.90, 0.17–0.94, and 0.15–0.93, respectively. Interspecies transferability values were 93.5%, 91.6%, and 91.1% for *G. echinata*, *G. glabra*, and *G. uralensis*, respectively. Phylogenetic analysis clustered cultivated (group 1) and wild (group 2) species into three and two subgroups, respectively. The reported markers represent a valuable resource for the genetic characteri z ation of *Glycyrrhiza* spp. for theanalysis of its genetic variability, and as a tool for licorice transferability. This is the first intraspecific study in a collection of *Glycyrrhiza* spp. germplasm using SSR markers.

## INTRODUCTION

The genus *Glycyrrhiza* L. (family Leguminosae) comprises approximately 20 recognized species distributed worldwide, most of which are perennial plants. The genus name *Glycyrrhiza* is derived from the ancient Greek word for 'sweet root' (Gr. *glykos* (sweet) + *rhiza* (root)), which was later Latinized to *liquiritia* and eventually to licorice (*Isbrucker & Burdock, 2006*).

Generally, licorice refers to the roots of *Glycyrrhiza glabra*, *Glycyrrhiza uralensis*, and *Glycyrrhiza inflata* (*Jung et al., 2015*). Licorice has been used as a medicinal herb to treat various diseases such as diabetes and depression. Glycyrrhizin, a triterpene found in the roots of *Glycyrrhiza* spp., exhibits pharmacological activities, including anticancer, detoxification, and anti-oxidant activities (*Montoro et al., 2011*). In addition, licorice root is a natural sweetener, and it is used as a flavoring agent in food production worldwide (*Snow, 1996*; *Gyawali et al., 2008*; *Zhang & Ye, 2009*). Licorice is sold in the form of slices or powder within Asian market, and therefore it is difficult to visually determine its variety and origin of production. Various studies have been conducted to identify varieties to secure their sovereignty at the national level in agreement with the Nagoya Protocol, an international treaty for sharing profits resulting from the utilization of biological resources (*Bang et al., 2011*; *Zhao et al., 2012*).

Several types of molecular markers utilize variations in DNA sequences, including random amplified polymorphic DNA, amplified fragment length polymorphism, single nucleotide polymorphism, and simple sequence repeat (SSR) markers. The genomes of plants and eukaryotic organisms contain a large number of SSRs (*Hamada, Petrino & Kakunaga, 1982*; *Delseny, Laroche & Penon, 1983*; *Tautz & Renz, 1984*), which are widely distributed in the coding and non-coding regions of nuclear and organellar DNA (*Vieira et al., 2016*). The SSR markers are co-dominant and have higher polymorphism and reproducibility than other DNA markers. Therefore, they are broadly used for identifying species and cultivars, as well as for analyzing genetic diversity and population structure (*Kalia et al., 2011*; *Ya et al., 2017*). Because SSR markers can be transferred between species (*Lichtenzveig et al., 2005*), they are suitable not only for determining genetic diversity but also for cross-species amplification, which reduces the cost and time associated with such analyses.

Wild species usually present higher resistance to abiotic and biotic stresses compared to cultivated plant species, and some lineages have desirable genetic traits. Genetic studies have been conducted on wild and cultivated species of crops such as potato and tomato (*Rick & Chetelat, 1995*; *Singh, Ocampo & Robertson, 1998*). Similarly, *Ashurmetov (1996)* examined the genetic relationships between wild and cultivated species of licorice. Recently, SSR markers were developed for licorice species based on the chloroplast genomes of *G. uralensis*, which is a cultivated species (*Liu et al., 2015*), and *G. lepidota*, which is a wild species (*Raveendar et al., 2017*; *Jo et al., 2018*); on the transcriptome of *G. uralensis* and *G. glabra*, also a cultivated species (*Um et al., 2016*); and on the nuclear genome of *G. lepidota* (*Lee et al., 2019*). However, to the best of our knowledge, molecular studies on genomic SSR markers of licorice have not been conducted. Thus, the present study

aimed to develop genomic SSR markers for licorice molecular genetic studies, including the differentiation between wild and cultivated species, their genetic diversity, and population structure.

## MATERIALS & METHODS

### Plant material and DNA isolation

The materials used in the present study comprised 11 accessions of *G. uralensis* and *G. glabra* obtained from the Ginseng Research Division at the National Institute of Horticultural and Herbal Science (NIHHS), South Korea, and 12 accessions obtained from the United States Department of Agriculture (USDA), including *G. uralensis*, *G. glabra*, *G. lepidota*, *G. echinata*, and *Glycyrrhiza* spp. (Table 1). Seedlings of each licorice accession were grown in pots of sterile soil in a greenhouse with three plants per accession. Leaf samples were purposively sampled from three plants per accession from 4-week-old seedlings and ground to powder using liquid nitrogen in a pestle–mortar; their genomic DNA was extracted using the Plant gDNA Extraction Kit (GeneAll, Seoul, Korea) following the manufacturer's protocol.

### Primer design and polymerase chain reaction

In our previous study, the chloroplast genome of *G. lepidota* (NCBI accession no. KY038482) was obtained by the *de novo* assembly of the low-coverage whole-genome sequence via a bioinformatics pipeline (http://phyzen.com) (*Raveendar et al., 2017*). We mined SSRs based on a whole-genome sequence that was not used in the chloroplast genome analysis of *G. lepidota*. The whole-genome sequence of *G. lepidota* were searched for SSRs using the MIcroSAtellite (MISA, http://pgrc.ipk-gatersleben.de/misa/) with the search criteria for minimum number of repeats set at six for mono-nucleotide repeats, five for di-nucleotide repeats, four for tri-nucleotide repeats, and three each for tetra-, penta-, hexa-, hepta-, octa-, nona- and deca-nucleotide motifs. Next, 100 SSR primer pairs with di- and tri-nucleotide repeat motifs were randomly designed using Primer 3.0 (http://primer3.sourceforge.net/) with the following criteria: primer length (18–23 bp, with optimum value 20 bp); Tm (54 °C–56 °C, with optimum value 55 °C); GC content (40%–60%, with the optimum value 50%); maximum Tm difference between forward and reverse primer 1.5 °C and product size range (100–350 bp with optimum value 250 bp). The first polymerase chain reaction (PCR) using all the primer sets was performed for four accessions of *G. lepidota* (CBG20-23) to establish the PCR conditions. The primers resulting in successful amplifications were then applied to assess the genetic diversity of the 23 *Glycyrrhiza* spp. accessions. The PCR mixture (total volume, 40 μL) contained 20 ng genomic DNA, 10 pmol each primer, 2.5 mM $MgCl_2$, 0.25 mM dNTPs, and 0.5 U Taq polymerase (Inclone, Deajeon, Korea). The amplification was performed on a CFX96 PCR detection system (Bio-Rad Laboratories, Hercules, CA, USA) and included 30 cycles of pre-denaturation at 94 °C for 5 min, denaturation at 94 °C for 30 s, annealing at 55–60 °C for 45 s, and extension at 72 °C for 1 min. The PCR products were separated and visualized using a Fragment Analyzer (Agilent Technologies, Santa Clara, CA, USA).

**Table 1** *Glycyrrhiza* **sp. accessions used in the present study.**

| Accession code | Species | Origin[a] | Institution[b] |
|---|---|---|---|
| CBG1 | *Glycyrrhiza uralensis* | CAN | NIHHS |
| CBG2 | *G. uralensis* | RUS | NIHHS |
| CBG3 | *Glycyrrhiza glabra* | CHN | NIHHS |
| CBG4 | *G. glabra* | CAN | NIHHS |
| CBG5 | *G. uralensis* | MNG | NIHHS |
| CBG6 | *G. glabra* | UZB | NIHHS |
| CBG7 | *G. uralensis* | CHN | NIHHS |
| CBG8 | *Glycyrrhiza* spp. | KOR | NIHHS |
| CBG9 | *Glycyrrhiza* spp. | KOR | NIHHS |
| CBG10 | *Glycyrrhiza* spp. | KOR | NIHHS |
| CBG11 | *Glycyrrhiza* spp. | KOR | NIHHS |
| CBG12 | *Glycyrrhiza echinata* | YUG | USDA ARS |
| CBG13 | *G. echinata* | DEU | USDA ARS |
| CBG14 | *G. uralensis* | KAZ | USDA ARS |
| CBG15 | *G. uralensis* | KGZ | USDA ARS |
| CBG16 | *G. glabra* | KGZ | USDA ARS |
| CBG17 | *G. glabra* | KGZ | USDA ARS |
| CBG18 | *Glycyrrhiza lepidota* | USA | USDA ARS |
| CBG19 | *G. lepidota* | USA | USDA ARS |
| CBG20 | *G. lepidota* | USA | USDA ARS |
| CBG21 | *G. lepidota* | USA | USDA ARS |
| CBG22 | *G. lepidota* | USA | USDA ARS |
| CBG23 | *G. lepidota* | USA | USDA ARS |

**Notes.**

[a] Refers to the region where the specimen was collected: CAN, Canada; RUS, Russia; CHN, China; MNG, Mongolia; UZB, Uzbekistan; KOR, Korea; YUG, Yugoslavia; DEU, Germany; KAZ, Kazakhstan; KGZ, Kyrgyzstan; USA, United States of America.

[b] Refers to the institution where the accessions are deposited: NIHHS, National Institute of Horticultural and Herbal Science; USDA ASR: United States Department of Agriculture Agricultural Research Service.

## Data analyses

The number of alleles, major allele frequency, genetic diversity, observed heterozygosity, and polymorphism information content (PIC) were analyzed using PowerMarker 3.25 (https://brcwebportal.cos.ncsu.edu/powermarker/). The rate of cross-amplification (transferability) among the 23 licorice accessions was measured using the following equation: Transferability (%) = amplicons (bands amplified by PCR) $\times$ 100/theoretical amplicons (primer number $\times$ sample size) (*Lee et al., 2015*; *Raveendar et al., 2015*). Phylogenetic analysis was performed using the Cavalli-Sforza chord distance (*Cavalli-Sforza & Edwards, 1967*) included in PowerMarker; the phylogenetic tree was constructed in MEGA4 (*Tamura et al., 2007*) using the unweighted pair group method with arithmetic mean.

**Table 2** Transferability of the 62 *Glycyrrhiza lepidota* simple sequence repeat (SSR) markers to other *Glycyrrhiza* species.

| Species (Sample size) | Theoretical amplicons (number) | Amplicons (number) | Transferability (%) |
|---|---|---|---|
| *Glycyrrhiza uralensis* (6) | 372 | 339 | 91.1 |
| *Glycyrrhiza glabra* (5) | 310 | 284 | 91.6 |
| *Glycyrrhiza echinata* (2) | 124 | 116 | 93.5 |
| *Mean* | | 268.6 | 92.07 |

## RESULTS

### Diversity of SSR markers and interspecies cross-amplification

From the whole-genome sequence of *G. lepidota*, 249,492 SSRs were mined. The frequency distribution of mined SSR repeat types is presented in Table S1. It was observed that mononucleotide repeats (218,320; 87.5%) were the most abundant repeat type, followed by dinucleotide repeats (11,056; 4.4%), tetra-nucleotide repeats (9,365; 3.8%), tri-nucleotide repeats (7580; 3%), penta-nucleotide repeats (3,170; 1.3%), and hexa-nucleotide repeats (1,387; 0.6%) in *G. lepidota* whole-genome, respectively (Table S1).

Sixty-two of the 100 selected primers amplified all the four accessions of *G. lepidota* (CBG20-23). Thus, these 62 primers were used to analyze the diversity of the 23 licorice accessions, which resulted in the identification of 549 alleles; the number of alleles per marker ranged from 2 (GL-gSSR-019) to 23 (GL-gSSR-028), with a mean of 9.6 alleles. The major allele frequency ranged from 0.11 (GL-gSSR-088) to 0.91 (GL-gSSR-019), with a mean of 0.349. The observed heterozygosity ranged from 0 (GL-gSSR-006, -019, -023, -068, -090, -097, and -100) to 0.70 (GL-gSSR-095), with a mean of 0.264. The maximum values of genetic diversity and PIC, indicating the genetic diversity of the markers, were 0.94 and 0.93, respectively, in GL-gSSR-028, whereas the minimum values were 0.17 and 0.15, respectively, in GL-gSSR-019. The mean values of genetic diversity and PIC were 0.760 and 0.730, respectively (Table S2). Cross-amplification analysis among licorice species revealed that *G. echinata* had the highest transferability (93.5%), followed by *G. glabra* (91.6%) and *G. uralensis* (91.1%); the mean value was 92.07% (Table 2).

### Phylogenetic analysis

The 23 accessions were classified into two clusters (Fig. 1), one corresponding to the cultivated plant species (group 1) and the other to the wild species (group 2). The 15 accessions within group 1 were arranged into three subgroups: group 1–1 contained two *G. glabra* accessions; group 1–2 contained six *G. uralensis* accessions, one *G. glabra* accession, and one *Glycyrrhiza* sp. accession; and group 1-3 contained two *G. glabra* accessions and three *Glycyrrhiza* sp. accessions. The seven accessions within group 2 were arranged into two subgroups, groups 2–1 and 2–2, which contained six *G. lepidota* and two *G. echinata* accessions, respectively.

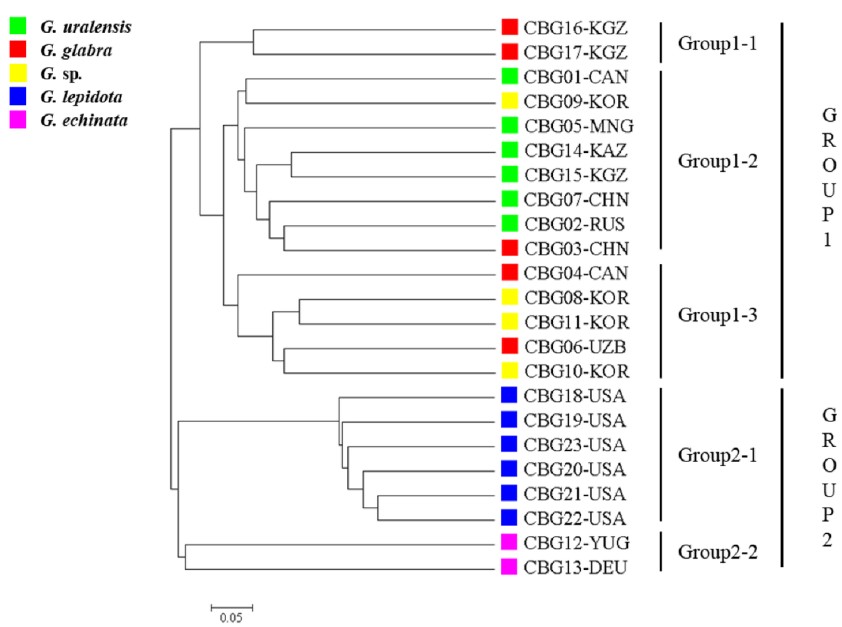

**Figure 1** Phylogenetic tree of the 23 *Glycyrrhiza* sp. accessions based on 62 simple sequence repeat markers developed here, as well as on the Cavalli-Sforza chord genetic distance.

## DISCUSSION

### Diversity of the SSR markers

The mean PIC value of the SSR markers developed in the present study using wild (*G. lepidota* and *G. echinata*) and cultivated (*G. uralensis* and *G. glabra*) licorice accessions was 0.730, which is higher than that reported for other medicinal plants: 0.314 for *Zingiber officinale* (*Pandotra et al., 2013*), 0.57 for *Codonopsis lanceolata* (*Kim et al., 2016*), and 0.272 for *Festuca arundinacea* (*Tehrani et al., 2009*). These results showed that the SSR markers developed in the present study have a higher diversity than those developed for other species. Moreover, *Um et al. (2016)* reported the PIC value of 0.56 for eight SSR markers based on *G. uralensis* genome using 22 accessions of cultivated licorice species (*G. uralensis* and *G. glabra*). To compare the markers developed in the present study with those previously reported, the PIC values of the 11 accessions of cultivated licorice species (*G. uralensis* and *G. glabra*) were measured, and a mean value of 0.624 was obtained. Thus, the SSR markers developed in the present study had a higher mean PIC value than those previously developed for *G. uralensis*, despite the lower number of accessions and higher number of primers used here. This might be attributed to the fact that the primers with PIC value higher than 0.63 accounted for 66.13% of the total primers. These results suggested that the markers developed in the present study should be more efficient for molecular genetic studies than those reported in previous studies due to their higher diversity.

In the diversity analysis based on repeat motifs, the mean PIC values of dinucleotide repeat motifs (32 sets of primers) and trinucleotide repeat motifs (30 sets of primers) were 0.796 and 0.688, respectively, showing a higher value for the dinucleotide repeat motifs. The

higher mean PIC value for dinucleotide repeats might be related to the fact that dinucleotide repeats are distributed throughout the genome, whereas trinucleotide repeats are mostly present in coding regions. Furthermore, trinucleotide repeats are under a relatively weaker selection pressure against mutations that alter the reading frame compared with the dinucleotide repeats, leading to a lower genetic diversity in the trinucleotide repeats (*Kalia et al., 2011*; *Vieira et al., 2016*). Considering that dinucleotide repeats had higher PIC values than trinucleotide repeats, despite the similar number of primers, the primers based on dinucleotide repeats are expected to be more useful for studies regarding diversity analysis.

## Interspecies cross-amplification

The mean interspecies transferability of licorice was 92.5%. The mean interspecies transferabilities of *Rubus coreanus* and *Allium sativum* were 73.52% (*Lee et al., 2015*) and 58.85% (*Lee et al., 2011*), respectively, and that of *Triticum aestivum* and *Secale cereale* were 48.4% and 35.3%, respectively (*Kuleung, Baenziger & Dweikat, 2004*). According to the report of *Erayman et al. (2014)*, the transferabilities of SSR markers developed for *Medicago truncatula*, *Phaseolus vulgaris*, and *Cicer arietinum* to *G. glabra*, *G. echinata*, and *G. flavescens* were 33% for *M. truncatula*, 11% for *P. vulgaris*, and 6% for *C. arietinum*. These results indicated that the markers developed for *Glycyrrhiza* accessions in the present study had higher transferability than those developed for other species. Interspecies crossing rates between *G. lepidota* and other licorice species were 51–75% (*Ashurmetov, 1996*), suggesting that crossing may occur under natural conditions, and genes of *G. lepidota* could affect other licorice species. We postulated that these are the likely reasons why interspecies transferability has become relatively high. In particular, *G. lepidota* had a high transferability with *G. echinata*, another wild species, possibly because these are more closely related to each other than to cultivated species. Such results showed that the SSR markers developed for the wild species *G. lepidota* can be efficiently used for other licorice species. However, as the number of resources used in the present study was smaller than that in other studies, future studies should include various species.

## Phylogenetic analysis of licorice accessions

In the present study, although CBG4 and CBG6 (*G. glabra* accessions) clustered with CBG16 and CBG17 (also *G. glabra* accessions), they were separated into groups 1–1 and 1–3, respectively. Such clustering could be due to the random selection of markers. In addition, the separation of *G. glabra* into two subgroups might be due to their genetic differences arising from their production origins. The four *Glycyrrhiza* spp. accessions CGB08, CGB09, CGB10, and CGB11 were separated into two groups, suggesting that CBG09 is genetically closer to *G. uralensis*, whereas CBG08, CGB10, and CGB11 are genetically closer to *G. glabra*. Accession CBG3 (*G. glabra*) clustered with *G. uralensis*, which might be due to either incomplete identification of resources during collection or insufficient numbers of licorice accessions (*Jo et al., 2018*; *Lee et al., 2019*). However, *Jo et al. (2018)* estimated that the wild species *G. echinata* was genetically close to the cultivated species *G. uralensis*.

## CONCLUSION

The clustering pattern obtained in the present study was congruent with the results of *Lee et al. (2019)*, indicating that the SSR markers that allow the analysis of intra-species diversity might also be efficient for interspecies differentiation. Moreover, the SSR markers developed in the present study might be successfully applied in molecular genetic studies aiming to differentiate wild and cultivated licorice and to determine their diversity.

## ACKNOWLEDGEMENTS

The authors thank Dr. Lee (Department of Herbal Crop Research, National Institute of Horticultural and Herbal Science, Rural Development Administration, Eumseong, Republic of Korea) for his assistance in collecting the licorice sample.

### Funding
The authors received no funding for this work.

### Competing Interests
The authors declare there are no competing interests.

### Author Contributions
- Jun Hyoung Bang conceived and designed the experiments, performed the experiments, analyzed the data, prepared figures and/or tables, authored or reviewed drafts of the paper, approved the final draft.
- Chi Eun Hong and Sebastin Raveendar conceived and designed the experiments, analyzed the data, contributed reagents/materials/analysis tools, approved the final draft.
- Kyong Hwan Bang, Kyung Ho Ma, Soon Wook Kwon and Hojin Ryu contributed reagents/materials/analysis tools, approved the final draft.
- Ick Hyun Jo and Jong-Wook Chung analyzed the data, contributed reagents/materials/-analysis tools, prepared figures and/or tables, authored or reviewed drafts of the paper, approved the final draft.

### Data Availability
Data is available at NCBI, accession numbers: KY038482.

### Supplemental Information
Supplemental information for this article can be found online at http://dx.doi.org/10.7717/peerj.7479#supplemental-information.

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
