# Peer review of "Development of genomic simple sequence repeat markers for Glycyrrhiza lepidota and cross-amplification of other Glycyrrhiza species"

_PeerJ, doi:10.7717/peerj.7479_

## Round 0.1 · original submission · Major Revisions

Dear Jong Wook,

Both reviewers have made suggestions to improve the manuscript. One of the reviewers is very worried about the process to identify and select the SSRs.

Please, consider the suggestions from reviewers and submit a new version of the manuscript if you think you could address them.

Cheers,
Marcial.

Reviewer 1 ·

Basic reporting

No comment

Experimental design

The method was not enough and not clearly descripted for SSR identification.

Validity of the findings

The development of SSR markers is one of the most important part for the manuscript. However, there was nothing mentioned for the identified SSR number, the frequency and distribution of different SSR types in the genome.

Additional comments

In this manuscript entitled "Development of genomic simple sequence repeat markers for Glycyrrhiza lepidota and cross-amplification of other Glycyrrhiza species" to describe the development of SSR markers in Glycyrrhiza lepidota, and use some of these markers to assess the genetic diversity. In general, this paper is good writing, but miss several important parts in methods and results. The overall research content was not enough for published on Peer J. Certain specific comments are addressed below:
Major comments
1. The development of SSR markers from the genomic sequence is an important component of the entire manuscript, while the detail on SSR analysis in the method and result was missed. For example, approximately 28,000 SSRs were identified in G. lepidota. What is the criteria used for identification of different SSR motif? What are the frequency, distribution and characterization of different SSR motif? These important information were not mentioned in the manuscript.
2. The Author Contributions was also missed in the manuscript.

Reviewer 2 ·

Basic reporting

This is a very well done work and it is very well presented. The authors have informed about a new set of markers very useful for different studies regarding this species.
In general, writing is very clear and the objectives are very well faced.
There are some minor things that I suggested to check before accepting this manuscript.

Experimental design

The experimental design and analysis follow the standard protocols for this kind of studies.
I feel that this work is original enough to be published here.
This a very well designed and performed work.

Validity of the findings

The authors have informed about the first set of SSR markers for this particular species. It is unique and hard to perform work.
Their findings, despite limited to this species or their related ones, will have an impact on the upcoming studies in different areas.

Additional comments

You have done a very intense and hard work. There are only some minor comments that I suggest you to consider.
You will find them in the PDF draft.

Annotated reviews are not available for download in order to protect the identity of reviewers who chose to remain anonymous.

---

## Round 0.2 · accepted · Accept

Dear Dr. Chung,

Congratulations! You study entitled "Development of genomic simple sequence repeat markers for Glycyrrhiza lepidota and cross-amplification of other Glycyrrhiza species" has been accepted.

Sincerely,
Marcial.

Reviewer 2 ·

Basic reporting

This the second revision of this manuscript. The authors made all the recommended changes and I suggest to accept the manuscript as it is now.

Experimental design

Minor changes recommended before were include.

Validity of the findings

No additional comments

Additional comments

No additional comments